# Ferromagnetism in Defected TMD (MoX_2_, X = S, Se) Monolayer and Its Sustainability under O_2_, O_3_, and H_2_O Gas Exposure: DFT Study

**DOI:** 10.3390/nano13101642

**Published:** 2023-05-15

**Authors:** Anjna Devi, Neha Dhiman, Narender Kumar, Wadha Alfalasi, Arun Kumar, P. K. Ahluwalia, Amarjeet Singh, Nacir Tit

**Affiliations:** 1Department of Physics, Himachal Pradesh University, Shimla 171005, India; 2Department of Physics, Swami Vivekanand Government College, Shimla-Kangra Rd, Ghumarwin 174021, India; 3Department of Physics, College of Science, United Arab Emirates University, Al-Ain P.O. Box 15551, United Arab Emirates; 4National Water and Energy Center, United Arab Emirates University, Al-Ain P.O. Box 15551, United Arab Emirates

**Keywords:** spin-polarized DFT, magnetic impurity and defect levels, ferromagnetism, electronic and magnetic properties, half-metallicity, spintronics, transition-metal di-chalcogenides

## Abstract

Spin-polarized density-functional theory (DFT) has been employed to study the effects of atmospheric gases on the electronic and magnetic properties of a defective transition-metal dichalcogenide (TMD) monolayer, MoX_2_ with X = S or Se. This study focuses on three single vacancies: (i) molybdenum “V_Mo_”; (ii) chalcogenide “V_X_”; and (iii) di-chalcogenide “V_X2_”. Five different samples of sizes ranging from 4 × 4 to 8 × 8 primitive cells (PCs) were considered in order to assess the effect of vacancy–vacancy interaction. The results showed that all defected samples were paramagnetic semiconductors, except in the case of V_Mo_ in MoSe_2_, which yielded a magnetic moment of 3.99 μ_B_ that was independent of the sample size. Moreover, the samples of MoSe_2_ with V_Mo_ and sizes of 4 × 4 and 5 × 5 PCs exhibited half-metallicity, where the spin-up state becomes conductive and is predominantly composed of $d_{xy}$ and $d_{z^{2}}$ orbital mixing attributed to Mo atoms located in the neighborhood of V_Mo_. The requirement for the establishment of half-metallicity is confirmed to be the provision of ferromagnetic-coupling (FMC) interactions between localized magnetic moments (such as V_Mo_). The critical distance for the existence of FMC is estimated to be $d_{c}≅$ 16 Å, which allows small sample sizes in MoSe_2_ to exhibit half-metallicity while the FMC represents the ground state. The adsorption of atmospheric gases (H_2_O, O_2_, O_3_) can drastically change the electronic and magnetic properties, for instance, it can demolish the half-metallicity characteristics. Hence, the maintenance of half-metallicity requires keeping the samples isolated from the atmosphere. We benchmarked our theoretical results with the available data in the literature throughout our study. The conditions that govern the appearance/disappearance of half-metallicity are of great relevance for spintronic device applications.

## 1. Introduction

Point defects in semiconductors can significantly affect their electronic structures, transport, and optical properties. They can either enhance or hamper the performance of the host material in device applications depending on their impact on the material’s properties. For instance, in gas-sensing applications, point defects can be useful in introducing dangling bonds on surfaces as intermediaries for capturing volatile organic compounds (VOCs) and toxic gas molecules [1,2,3,4]. On the other hand, the defects can also have a negative impact on the transport and the optical properties such as trapping centers or DX-centers in semiconductors [5]. These kinds of defects are well-characterized by deep-level transient spectroscopy (DLTS) [6].

In the characterization of defects in TMD ML, Conzalez and coworkers [7] used both scanning tunneling microscopy (STM) and Keldysh non-equilibrium Green-function formalism within density functional theory (DFT) to model the images in characterizing the defects on a free-standing MoS_2_ ML. They showed that the most abundant defects were S and Mo vacancies, as well as S_Mo_ and Mo_S_ anti-sites. In a related work, Kc and coworkers [8] used STM and DFT to study point defects in MoS_2_ MLs. They reported that V_S_ is more energetically favorable than V_Mo_ and thus more abundant. The research group of Liu [9] conducted a similar experimental study using STM and scanning tunneling spectroscopy (STS) on MoSe_2_ bilayer and monolayer systems. They showed that the prominent point defects were Mo and Se vacancies and anti-sites.

In simulating the effects of defects on the electronic characteristics, Shafqat et al. [10] used the Amsterdam density functional (ADF) to study defects in MoSe_2_ ML. They showed two kinds of defects (i.e., anti-site Mo_Se2_ and Mo vacancies, V_Mo_) responsible for introducing magnetism into the system. Ma and coworkers [11] used DFT to study the effects of vacancy defects on the magnetic properties of TMD MLs. They used a supercell with a size of 4 × 4 primitive cells (PCs) to study H-adsorbed (MoSe_2_, MoTe_2_, WS_2_) MLs and F-adsorbed (WS_2_, MoSe_2_) MLs and showed the existence of long-range antiferromagnetic coupling between local moments up to a distance of about d_c_ ~ 12 Å. In a related computational work, Yang and coworkers [12] applied spin-polarized DFT to a 5 × 5 PC sample of WSe_2_ ML to study the effect of simple and complex vacancies on the electronic and magnetic properties. These authors showed that only V_W2_ and V_WSe6_ can introduce magnetism into WSe_2_ ML with magnetic moments of 2 μ_B_ and 6 μ_B_, respectively. The magnetic moments are attributed to the atoms around the vacancies, and, in particular, WSe_2_ with V_W2_ was reported to be half-metallic. It is worth mentioning that half-metallicity ought to be the primary distinguishing feature in the accommodating material for spintronic applications [13,14].

From the perspective of stability under ambient conditions, defect-free TMDs have shown long-range structural and optical stability in ambient conditions due to their inert surfaces and the absence of dangling bonds [15]. However, a thorough investigation of the literature reveals that TMDs can undergo environmental degradation and transformation, which is significant for managing development risks [16]. Pető, János, et al. [17] reported the spontaneous replacement of S atoms by oxygen atoms (one by one) in a mechanically exfoliated MoS_2_ ML on long-range exposure to O_2,_ producing 2D MoS_2−*x*_O*_x_* crystals. Nurdiwijayanto et al. [18] synthesized MoS_2_ ML using Li-intercalation and stored it in ambient air. They detected degradation in the sample with oxidation after 2 months of exposure, which can be prevented if the material is stored in an inert atmosphere. In another study conducted by Gioele Mirabelli and coworkers, [19] the stabilities of five different TMDs (i.e., MoS_2_, MoSe_2_, MoTe_2_, HfS_2_, and HfSe_2_) were studied. They reported that HfSe_2_ was the least stable under ambient conditions, showing signs of degradation after 24 h due to the transformation into HfO_2_. On the other hand, MoS_2_ was the most stable sample during the study period and, consequently, can be used for electronic applications. These experiments revealed a general tendency toward the deteriorating stability of TMD MLs under ambient conditions, particularly when the chalcogenide element shifts from S to Se and then Te (i.e., although MoS_2_ is very stable, MoSe_2_ started oxidizing on the ninth day and MoTe_2_ was the most reactive among the Mo-based TMDs).

## 2. Computational Method

Two computational methods were used to ensure the highest authenticity in our observations. The first one was based on the Spanish initiative for electronic simulations with thousand atoms (SIESTA), [20] whose efficiency stems from the use of a localized basis set to deal with large systems, and its accuracy can be competitive with plane-wave ab initio methods by selecting appropriate pseudo-potentials and exploring high K-mesh grids. The second method is a widely popular and reliable method based on the Vienna Ab Initio Simulation Package (VASP), [21] which is used to benchmark our results and deal with more refined calculations such as in the case of the existence of half-metallicity. Furthermore, group theoretical analysis is involved in analyzing the splitting of states near the Fermi level to contrast the effect of the defect and trace the origins of half-metallicity in cases where it exists.

The density functional theory (DFT), which is common to SIESTA and VASP, employs the general gradient approximations (GGA) with the Perdew–Burke–Ernzerhof (PBE) [22] functional for the description of the exchange–correlation interaction between electrons. The basis sets of the Hamiltonians in the two methods are different, as mentioned above (i.e., SIESTA’s basis set is based on localized orbitals, whereas VASP’s basis set is based on plane waves). For the SIESTA method, we used a double zeta polarization (DZP) function basis set and a constant energy cutoff of *E_cut_* = 300 Ry. The Brillouin zone integration was carried out using the Monkhorst–Pack technique [23], with a K-mesh size of 10 × 10 × 1. The criteria for convergence for the total energy and force per atom were set at 10^−6^ eV and 1 meV/Å, respectively. On the other hand, for the VASP package, the projector-augmented plane-wave (PAW) method was utilized. PAW pseudo-potential with non-local projectors for the molybdenum (Mo) 4s, 4p, 4d, and 5s atomic orbitals and the sulfur (S) 3s and 3p orbitals were included. The spin-orbit coupling (SOC) was self-consistently included [24]. An energy cutoff of *E_cut_* = 400 eV, with a K-mesh size of 5 × 5 × 1 in the Monkhorst–Pack scheme, was implemented in the geometry optimization with a tolerance of the total energy and force of about 0.01 eV/Å and 10^−5^ eV, respectively. For both the SIESTA and VASP methods, a default value for the on-site U Hubbard parameter of U = 4.5 eV was used, which was almost the default value and in good agreement with the adjusted values calculated by Mann et al. [25]. The same value was also used in our previous work [26].

The binding energy ($E_{b}$), which is equivalent to the average cohesive energy per atom, of MoX_2_ (where X = S, Se) with and without vacancy was calculated using the formula:(1)$$E_{b}=\frac{E_{tot}-(mE_{Mo}+nE_{X})}{n+m}$$
where $E_{tot}$ is the total energy of the system, $E_{Mo}$ and $E_{X}$ are the energies of the isolated atoms of Mo and chalcogenide, and m and n are the numbers of Mo and X atoms in the system, respectively.

The formation energy of vacancy ($E_{f}$), for instance, for V_Mo_ in MoS_2_, is defined by:(2)$$E_{f}=E_{tot}^{Bulk}-E_{tot}^{Mo}-E_{tot}^{w/Vac}$$
where $E_{tot}^{Bulk}$, $E_{tot}^{Mo}$, and $E_{tot}^{w/Vac}$ represent the total energies of the bulk sample, isolated Mo atom, and sample with the Mo vacancy, respectively. If $E_{f}>0$, the formation of the vacancy is endothermic, and if $E_{f}<0$, the formation of the vacancy is exothermic.

The adsorption energy ($E_{ads}$) of the gas molecule on the adsorbent substrate is defined as:(3)$$E_{ads}=E_{tot}^{Sub/gas}-E_{tot}^{Sub}-E_{tot}^{gas}$$
where $E_{tot}^{Sub/gas}$, $E_{tot}^{Sub}$, and $E_{tot}^{gas}$ represent the total energies of the substrate with and without the gas molecule and the isolated gas molecule. It is worth mentioning that we had to carry out the spin-polarized DFT calculations, as the transition metal (Mo) atoms are included in our TMD systems.

## 3. Results and Discussion

### 3.1. Atomic Relaxations

Atomic relaxations were performed using the SIESTA code on supercells (SCell) of MoX_2_, where X represents a chalcogenide atom, either S or Se, in its monolayer form, comprising pristine and single-vacancy-defected cases. To simultaneously preserve the validity of translational symmetry (i.e., Bloch theorem) and assess the effect of the single vacancy defect on the electronic and magnetic properties, we selected the supercells’ sizes: 4 × 4, 5 × 5, 6 × 6, 7 × 7, and 8 × 8 primitive cells. Three types of point defects were incorporated into each of these samples. Figure 1 displays the energy-optimized structures corresponding to the case of an SCell of MoS_2_ with a size of 5 × 5 primitive cells after the achievement of full atomic relaxations, with the following starting configurations: (a) Pristine, (b) Mo vacancy “V_Mo_”, (c) S vacancy “V_S_”, and (d) S_2_ di-vacancy “V_S2_”. Some relevant geometry data of the relaxed structures of MoS_2_ and MoSe_2_ in both the pristine state and with the defects mentioned above are summarized in Appendix A, respectively. For instance, these tables show the Mo-X bond lengths and vacancy contents for all the studied samples, along with the corresponding average formation or binding energy for each configuration.

For the sake of benchmarking our calculation of formation energy, as indicated in Appendix A, the results of the average binding energies are shown for the pristine and defected MoS_2_ and MoSe_2_ MLs, respectively. In the case of pristine MoS_2_ ML, we found that the average binding energy (i.e., cohesive energy per atom) was $E_{bind}$ = −4.915 eV/atom, which is in good agreement with both the ab initio calculations of Ding and coworkers [27] (−5.00 eV/atom) using pseudo-atomic numerical orbitals and our previous work [28] (−4.890 eV/atom) using Troullier–Martins’ norm-conserving relativistic pseudopotentials. In the case of pristine MoSe_2_ ML, our result was $E_{bind}$ = −4.401 eV/atom, which is also in good agreement with the latter references (−4.530 eV/atom and −4.401 eV/atom, respectively). It should be emphasized that the greater the charge transfer from the metal layer to the chalcogenide layer, the higher the binding (or cohesive) energy. This is consistent with what is experimentally well-established, that is, MoS_2_ is considered the most thermodynamically stable.

#### 3.1.1. MoS_2_ with Vacancy

From the pristine structure perspective, the metal atom “Mo” had a coordination of 6 and the chalcogenide atom “S” had a coordination of 3. Concerning the first type of vacancies, the removal of one Mo atom (creation of V_Mo_) left the six sulfur atoms close to V_Mo_ with just two neighbors each. The atomic reconstructions led to a decrease in Mo-S bond lengths but an increase in S-S separation, which increased from d_S-S_ = 3.21 Å to 3.31 Å after the removal of the Mo atom. This increase in the S-S separation displayed an *outward relaxation* and can be attributed to the increase in electrostatic repulsion between two similarly charged atoms with high electronegativity [29]. It should be emphasized that the outward relaxation of S-S’s first nearest neighbors (FNN) was larger than that of S-S’s second nearest neighbors (SNN) [29].

Regarding the second type of vacancy, the removal of the S atom left the three neighboring Mo atoms with a coordination of less than 5. The average nearest neighbor Mo-Mo distance in the pristine MoS_2_ monolayer was d_Mo-Mo_ = 3.21 Å but it decreased to about 3.12 Å after the removal of the S atom. This decrease in the Mo-Mo distance in the neighborhood of the V_S_ is indicative of an inward relaxation. In the case of the V_S_, the inward relaxation of the SNN atoms was larger than that of the FNN atoms, likely due to the decrease in electrostatic repulsion between the Mo atoms [30].

The third type of defect, V_S2_, further decreased the coordination of its three neighboring Mo atoms to four, rather than six in the case of the pristine state. In Appendix A, it can be seen that the bond length increased with respect to the V_S_, which can be attributed to the increase in the inward relaxations of Mo in the vicinity of the V_S2_. The d_Mo-Mo_ increased to 2.87 Å compared to 3.21 Å for the pristine monolayer.

#### 3.1.2. MoSe_2_ with Vacancy

In contrast to the previous study of MoS_2_ material, the Mo vacancy in MoSe_2_ resulted in the inward relaxation of the six Se atoms neighboring the V_Mo_. The distance d_Se-Se_ = 3.34 Å in the pristine MoSe_2_ decreased to 3.27 Å in the case of V_Mo_. Nevertheless, the single vacancy of Se persisted, causing an inward relaxation of the three Mo atoms neighboring the V_Se_ because the d_Mo-Mo_ decreased to 3.18 Å. This distance decreased even further to 2.85 Å in the case of the V_S2,_ exhibiting a consistent trend and confirming more inward relaxations.

Appendix A display the average binding energy, which should be an indicator of the stability of the structure. As a reference, the average binding energies for MoS_2_ ML and MoSe_2_ ML were found to be 4.915 eV/atom and 4.401 eV/atom, which are in good agreement with our previous studies [31]. The results of the absolute value of binding energy versus the sample size are displayed in Appendix A for both (a) MoS_2_ and (b) MoSe_2_. The common trend for the strength of the binding energies was as follows: $\left|E_{b}\left(pristine\right)\right|>\left|E_{b}(V_{X})\right|>\left|E_{b}(V_{X2})\right|>\left|E_{b}(V_{Mo})\right|$ and was likely due to the size of the missing atom(s). Furthermore, the magnitude of the binding energy increased with the sample size as the system approached the bulk structure. In the case of MoSe_2_ with V_Se_ and V_Se2_ defects, the systems had E_b_ converging fast to the bulk value (see, for instance, the samples of 7 × 7 and 8 × 8 PCs, which almost restored the binding energy of the bulk). Appendix A shows that MoX_2_ with V_X_ and V_X2_ was more thermodynamically stable than MoX_2_ with V_Mo_.

Our results of the formation energy of the V_Mo_ single vacancy in MoS_2_ $E_{f}$ = −1.341 eV are also in good agreement with the results of the ab initio calculations of Ding and coworkers [27] ($E_{f}$ = −1.420 eV). Moreover, in the case of the VMo single vacancy in MoSe_2,_ $E_{f}$ = −1.341 eV was also close to the value found by Ding et al. [27] ($E_{f}$ = −1.210 eV). As mentioned in Section 2, when $E_{f}<0$, the formation of such a vacancy requires an exothermic reaction. Our results are not only consistent with those in the literature but also reveal the thermodynamic stability of such defects.

### 3.2. Electronic Structures

Spin-polarized calculations were carried out on all the samples previously relaxed to probe both the band structures and the total density of states (TDOS). Figure 2 and Appendix A show these results against sample sizes of 4 × 4, 5 × 5, 6 × 6, and 8 × 8 primitive cells and with the following point defects: (a) V_Mo_, (b) V_X_, and (c) V_X2_ in TMD of the MoX_2_ monolayer, with X = S and Se, respectively. The in-band structures and spin-up and spin-down bands are represented by black and red curves, respectively. The Fermi level is used as an energy reference (i.e., E_F_ = 0) and is represented by horizontal red dashes. By looking at Appendix A, one can deduce that the defected MoS_2_ ML is always paramagnetic, regardless of the existing vacancy defect because there is no distinction between spin-up and spin-down in the bands and TDOS plots. Similarly, the same observation can be made for MoSe_2_ with chalcogenide vacancies (i.e., V_Se_ and V_Se2_). The only case of V_Mo_ in MoSe_2_ was found to produce magnetism and yield magnetization in the samples, irrespective of their sizes.

#### 3.2.1. Effect of Vacancies on Bandgap Energy

By looking at the case of MoS_2_, as shown in Appendix A, it can be seen that the pristine form exhibited a direct band gap at the K-point of the Brillouin zone with a value of E_g_ = 1.626 eV. This value is in good agreement with the results of other DFT methods [31,32] that reported values of 1.72 eV, 1.80 eV, and 1.70 eV, respectively, and is slightly lower than the experimental value of 1.80 eV reported by Boker and coworkers [31] using angle-resolved photoelectron spectroscopy (ARPES). It is expected that DFT will underestimate the bandgap energy.

Both pristine and vacancy-defected MoS_2_ were shown to be paramagnetic, as both spins were degenerate and spin-orbit coupling had a negligible effect on the band structures, as demonstrated in Figure 2. The bandgap energy versus the sample size is displayed in Appendix A and shows that the bandgap energy decreased to 0.2 eV, 1.0 eV, and 1.1 eV after introducing the vacancies V_Mo_, V_S2,_ and V_S_, respectively. It seems that the dangling bonds formed in the vicinity of the vacancy and originating on the neighboring atoms introduced localized gap states. The charge states and characteristics of these gap states are examined in the next sub-section. In brief, the three types of vacancies in MoS_2_ maintained the semiconducting properties of the material but also introduced some deep-level “trap” states in the bandgap.

The spin-polarized electronic structures of vacancy-defected MoSe_2_ MLs are presented in Figure 2. In pristine form, MoSe_2_ ML has direct bandgap energy at the K-point in the Brillouin zone with a value of E_g_ = 1.460 eV (see Appendix A). This value is in good agreement with the theoretical results reported in the literature, for example, by Ma and coworkers [11] and Liu and coworkers, [33] who reported a similar value of 1.44 eV. Meanwhile, our bandgap value was very close to the experimental value of Ugeda and coworkers [34], who reported 1.61 ± 0.11 eV using the scanning tunneling spectroscopy (STS) technique, and the value of 1.66 eV reported by Ross and coworkers [35] using photoluminescence (PL) spectroscopy.

The introduction of a chalcogenide atomic vacancy or di-vacancy (i.e., V_Se_ and V_Se2_) did not introduce any magnetism into the system, irrespective of the sample size. The only vacancy that had the ability to introduce magnetism into the system was the molybdenum vacancy “V_Mo_”. Appendix A displays the bandgap energy for each spin in the three cases of vacancies in the MoSe_2_ monolayers versus the sample sizes. Appendix A shows that the effects of both V_Se_ and V_Se2_ were very similar in introducing no magnetism but only deep donor states in the gap at about 1.0 eV (i.e., at about E_C_ − 0.46 eV). The lowest bandgap energy corresponded to a V_Mo_ defect. Furthermore, it is remarkable that the V_Mo_ introduced magnetization into MoSe_2_ with a magnetic moment of about M ≈ 4 μ_B_, which can be attributed to the absence of four electrons in the d shell of the Mo vacancy and the localization of the wave function on the FNN chalcogenide atoms. In addition to the formation of the magnetic moment, the results of the band structures revealed the occurrence of half-metallicity in just two samples with respective sizes of 4 × 4 and 5 × 5 primitive cells, as shown in Appendix A. Although spin-down exhibited semiconducting behavior with a bandgap energy of about 0.708–0.747 eV in the latter samples, the spin-up exhibited a metallic transition by having few bands crossing the Fermi level. Half-metallicity was also reported by Ma et al. [11] in 2011 on a sample of 4 × 4 PCs of MoSe_2_ ML with V_Mo_. Because of its importance in spintronic device applications, the origins of the half-metallicity behavior should be further investigated (see the sub-sections below).

#### 3.2.2. Type of Gap States Due to Vacancies

Defects, specifically vacancies, result in the formation of dangling bonds, which usually introduce gap states. Furthermore, it is known that in TMD, the metal atoms predominantly contribute to the electronic structures of the conduction-band minimum (CBM) and the valence-band maximum (VBM) [29]. As can be observed in Figure 3 and Appendix A, the existence of vacancies can introduce energy levels near the Fermi energy (within the bandgap) and can shift the VBM and CBM with respect to the vacuum level or the original band edges of the pristine state. The shifts of the VBM and CBM are more pronounced in the case of the V_Mo_ than in any other case.

Appendix A displays the gap states obtained due to the various vacancies in (a) MoS_2_ ML, and (b) MoSe_2_ ML. The Fermi level was chosen as an energy reference (E_F_ = 0) and was maintained for all the samples. It is represented by a horizontal dashed red line. The VBM and CBM of the pristine state are represented by horizontal green dashed lines. No spin was included in the case of MoS_2,_ with and without defects, as the system was paramagnetic. Spin was included only in the case of V_Mo_-defected MoSe_2,_ as the system became ferromagnetic. Consistent with Appendix A, the energy levels corresponding to V_Mo_ were always the lowest in energy and were located close to the Fermi level with respect to the VBM, as shown in Appendix A. By using the Fermi level as an energy reference in Appendix A, for MoS_2_ (Appendix A), the defect states corresponding to V_S_ and V_S2_ were located above E_F_, whereas those of V_Mo_ were located below E_F_. This indicates that the dangling bonds of V_Mo_ were negatively charged, whereas those of V_S_ and V_S2_ were empty of charges. The gap states due to the vacancies in MoS_2_ ML were previously studied on a small sample of 4 × 4 PCs by Feng and coworkers [36].

Concerning the case of MoSe_2_ presented in Appendix A, the energy levels due to the V_Se_ and V_Se2_ seemed to be located at or slightly above the CBM, making resonance states at the conduction band. The Mo vacancy was more interesting as it yielded not only magnetism but also half-metallicity in MoSe_2_. The energy levels corresponding to the spin-down states were always located above the Fermi level at an energy of about E ≈ E_F_ + 0.2 eV, thus exhibiting semiconducting behavior. However, the energy levels corresponding to the spin-up states were located just near the Fermi level (i.e., in the two cases of the 4 × 4 and 5 × 5 PC samples, the gap states due to the defects met the Fermi level but in the other three 6 × 6, 7 × 7, and 8 × 8 PC samples, they were located slightly below the Fermi level). Hence, two samples were characterized to produce the half-metallicity characteristic. This is further investigated in the next sub-section, where the behaviors of the spin-polarized eigenfunctions of HUMO/LUMO and the Fermi-level states (of both spins) and the orbital density of states (ODOS) are studied.

### 3.3. Magnetic Properties

Spin-polarized band structure calculations confirmed that all the studied systems were paramagnetic, except for one case that corresponded to the Mo vacancy in MoSe_2_ ML. The calculation of the magnetic moment, as shown in Appendix A, showed that M = 3.99 μ_B_, irrespective of the sample size. This value is comparable to the value of 3.27 μ_B_ reported by Ma and coworkers [11] using VASP on a small sample size of 4 × 4 PCs. Figure 3 displays the spin profiles (magnetic moment or spin-vector distribution) in the cases of samples 4 × 4, 5 × 5, 6 × 6, 7 × 7, and 8 × 8 PCs, which can be used to assess the origin of the orbitals/species contributing to the magnetization in MoSe_2_ with a Mo vacancy. Figure 3 shows that the spin vectors of the atoms in the vicinity of the vacancy were aligned parallel to the same direction (i.e., indicating the formation of “ferromagnetism”). The spin distribution was localized on sites neighboring the Mo vacancy and was independent of the sample size. Different species of atoms (i.e., Se and Mo atoms) made different contributions to the magnetic moments. Se atoms, which were closer to the vacancy, made a greater contribution to the magnetization through the dangling bonds. In addition, there were six Se atoms consisting of the FNNs of V_Mo_, each contributing 0.414 μ_B_, and six Mo atoms consisting of the SNNs of V_Mo_, each contributing 0.212 μ_B_. Hence, the total magnetic moment due to the FNN and SNN contributions was 3.838 μ_B_ (i.e., M = 6 × 0.414 + 6 × 0.212 = 3.838 μ_B_). The rest of the atoms around the vacancy contributed very little to the magnetization.

### 3.4. Origin of Half-Metallicity

Half-metallicity is a collective physical phenomenon that cannot take place unless the system is ferromagnetic. Usually, the starting point is that both spins behave as semiconducting and then under certain conditions, one of them becomes conducting. Or vice versa, initially, the two spins are metallic, and if some physical conditions are provided, one spin changes character to become semiconducting. In our present case, both spins had a semiconducting character, but with the incorporation of a Mo vacancy into just two samples of MoSe_2_ (sizes 4 × 4 and 5 × 5 PCs), the spin-up became conducting and the system became half-metallic. The coexistence of two characteristics (i.e., metallic and semiconducting) for the two independent spins can be attributed to the molar content of the Mo vacancy. The main requirement for half-metallicity is that the compromised content of V_Mo_ cannot be too low or too high. This may be due to an interaction between the magnetic moments of the V_Mo_ and its mirror vacancies, as the Hamiltonian is constructed to preserve the periodic boundary conditions. To investigate the half-metallic character in the MoSe_2_ samples with V_Mo_ of sizes 4 × 4 and 5 × 5 PCs, it is necessary to examine the eigenwave functions of the spin-up states at the Fermi level and both the HOMO and LUMO states of the spin-down states, as well as the orbital density of states (ODOS) for three sample sizes (i.e., 4 × 4, 5 × 5, and 6 × 6 PCs) to better understand the metallization of the spin-up state when the density of V_Mo_ becomes moderately high, particularly in the small samples of sizes 4 × 4 and 5 × 5 PCs.

According to the point-group theoretical analysis, [29] in trigonal prismatic symmetry, the five d states of the metal atom in TMD (e.g., Mo atom) should split into three sub-groups: (i) a singlet A_1_′ consisting of the $d_{z^{2}}$ orbital, (ii) a doublet E’ consisting of the $d_{xy}$ and $d_{x^{2}-y^{2}}$ orbitals, and (iii) a doublet E” consisting of the $d_{yz}$ and $d_{zx}$ orbitals. With this in mind, Figure 4 and Figure 5 should be discussed simultaneously. Figure 4 displays the results of the PDOS and ODOS for two samples of sizes 4 × 4 and 5 × 5 PCs. The PDOS plots (in Figure 4a) show that the V_Mo_ resulted in dangling bonds on the neighboring six Se atoms (i.e., two gap states of the spin-up and one gap state of the spin-down). Both the PDOS and ODOS (Figure 4a,b) show that the spin-up at the Fermi level was mainly due to the contributions of the $d_{z^{2}}$ and $d_{xy}$ orbitals of the Mo atoms and the s and p orbitals of the Se atoms close to the V_Mo_ vacancy. This is consistent with the group theory, which predicted that the preceding d states would be close to the Fermi level (i.e., comprising the pristine HOMO/LUMO states, respectively); therefore, the magnetic interaction between the vacancy and its mirror may be the reason for mixing the two d states and producing the metallization of the spin-up states. One can further see that the second gap state of the spin-up located below the Fermi level was mainly due to the contributions of the Se atoms and thus can be solely attributed to the dangling bonds attached to the Se atoms in the vicinity of the vacancy. The third gap state, located slightly higher than the Fermi level in energy, is considered the LUMO state of the spin-down states and was clearly localized on the (Mo and Se) atoms in the vicinity of the vacancy.

Figure 5 displays the eigenfunction plots of the HOMO/LUMO states of the spin-down and the Fermi states of the spin-up. The figure corroborates our discussions of the PDOS and ODOS. For both samples of sizes 4 × 4 and 5 × 5 PCS, regarding the spin-down states, the HOMO was mainly due to the contributions of both the $d_{z^{2}}$ and $d_{xy}$ orbitals of the Mo atoms, whereas the LUMO was localized on the Se atoms in the vicinity of the vacancy. On the other hand, for the spin-up states at the Fermi level, the eigenstate can be attributed to the contributions of the $d_{z^{2}}$ and $d_{xy}$ orbitals of the Mo atoms, located in the vicinity of the V_Mo_. In Figure 5, for the samples of sizes 6 × 6 and 8 × 8 PCs, the disappearance of half-metallicity can be attributed to the decoupling of the vacancy–vacancy interaction as the distance between them increased beyond the critical coherence length of ferromagnetic coupling (ferromagnetic coupling was estimated to be approximately d_c_ ≈ 12 Å by Ma et al. [11]). Consequently, relaxations of both the spin-up and spin-down states were carried out for the SNN and FNN atoms near the V_Mo_ in the 6 × 6 and 8 × 8 samples. This is consistent with Figure 3, which shows the robustness of spin localization near the defect to maintain a constant magnetization. Additionally, it reveals the ferromagnetic decoupling between the Mo vacancies and the loss of half-metallic character in the larger samples.

### 3.5. Anti-Ferromagnetism versus Ferromagnetism

Usually, the magnetic interaction between two localized magnetic moments is anti-ferromagnetic in order to form the ground state. However, this is only the case when the localized moments are in close proximity. For instance, in our recent work [37], we carried out a comparative assessment of a double-atom catalyst consisting of manganese Mn_2_ embedded in a large pore of C_2_N by comparing the anti-ferromagnetic (AFM) and ferromagnetic (FM) states. The results confirmed that the AFM state was the ground state, as it had a total energy lower than the FM state (i.e., $∆E=E_{AFM}-E_{FM}=-0.595\;eV$). Furthermore, the bond length of the Mn_2_ dimer in the C_2_N pore was shorter in the AFM state (i.e., $d_{Mn-Mn}^{\left(AFM\right)}=2.178$ Å > $d_{Mn-Mn}^{\left(FM\right)}=2.207$ Å [37]. So, the impact of our previous work on the current investigation is that one should question what would happen when the vacancies physically exist in one sample and possibly interact via AFM coupling, particularly if the AFM state is the ground state and the FM state is the excited state. In this case, what would the effect be on half-metallicity? So, here we used a large sample of MoSe_2_ consisting of 8 × 8 PCs, and four vacancies of Mo (i.e., V_Mo_) were included in a uniform distribution by repeating the sample of 4 × 4 PCs, with a single vacancy, V_Mo_, in both *xy* directions, as shown in Figure 6a. The new sample was four times the size of the original 4 × 4 PC sample. We assessed both the AFM and FM states on this large new sample. The results showed that the atomic relaxation always led to an AFM state, which represented the ground state. This trend might be due to the fact that the distance between the magnetic impurities was so large (i.e., $d_{Mn-Mn}^{}=13.20$ Å), which favored the FM state as the ground state over the FM state. Figure 6b shows the spin-polarized band structure and TDOS, which confirm the existence of the half-metallic character. Figure 6b shows that the bands of the spin-up states (in orange) were metallic, dispersive, and crossed the Fermi level, whereas the bands of the spin-down states (in black) remained semiconducting. Moreover, the spin vectors, as shown in the side-view of Figure 6a, were localized within the vacancies. This confirms that the ground state was ferromagnetic and the material maintained its half-metallic behavior. In other words, half-metallicity requires the provision of a magnetic–coupling interaction of intermediate strength at the level of ferromagnetic coupling.

### 3.6. Environmental Effects

It is necessary to assess the environmental effects on the properties of TMD (MoX_2_, X = S, Se) MLs, especially the half-metallic character. We mainly focused on atmospheric gases that are oxidizing (e.g., O_2_, H_2_O, and O_3_) to study their adsorption properties on our current samples of MoX_2_ (X = S, Se) monolayers. Figure 7 shows the DFT results of the atomic relaxations of these three gases on the four samples: (a) MoS_2_:V_Mo_ (i.e., on a Mo vacancy); (b) MoS_2_:V_S_ (i.e., on a S vacancy); (c) MoSe_2_:V_Mo_ (i.e., on a Mo vacancy); and (d) MoSe_2_:V_Se_ (i.e., on a Se vacancy). The computational supercell contained 4 × 4 PCs, with the inclusion of one vacancy. The gas molecule was initially placed on the top of the vacancy within a distance of about 3.0 Å to start the process of atomic relaxation. The results of the atomic relaxations, which are summarized in Table 1, comprise the adsorption energy, charge transferred with the molecule, and the closest distance between the molecule and substrate (adsorbent bed). The results of the atomic relaxations can be summarized as follows: (i) The common trend among all gas adsorption processes was the weak physisorption exhibited by O_2_ molecules on all four samples. The adsorption energy reported in Table 1 was small and within the range of about [−0.18, −0.10] eV. (ii) Ozone molecule O_3_ seemed to be the strongest and interacted with all samples through either a chemisorption associated with a dissociation (e.g., see Figure 7a,c,d) or a strong physisorption (e.g., see Figure 7b). (iii) The vapor molecule H_2_O seemed to exhibit weak physisorption on most of the samples (e.g., see Figure 7a,c,d) but it exhibited chemisorption associated with dissociation on one sample, MoS_2_:V_S_ (e.g., Figure 7b).

The spin-polarized total density of states (TDOS) corresponding to the four studied samples before and after the adsorption of the three gas molecules (i.e., H_2_O, O_2_, O_3_) are displayed in Figure 8. The results of the bandgap energy and magnetization are summarized in Table 2. The results of the TDOS versus the samples under the effect of gas adsorption can be summarized as follows: (a) Sample MoS_2_:V_Mo_: The adsorbent bed was paramagnetic with Eg = 0.234 eV, as shown in Figure 8a. Only the oxidation of O_2_ can change the magnetic state of the substrate, even though the interaction with that molecule is a kind of weak physisorption. The change in magnetization due to O_2_ physisorption was reported to be approximately ΔM = 2.0 μ_B_ and can be attributed to an induced magnetization in the localized molecular orbitals. (b) Sample MoS_2_:V_S_: The adsorbent was paramagnetic with null magnetization and Eg = 1.10 eV, as shown in Figure 8b. The effect of humidity (H_2_O molecule) seemed to be significant, as it reduced the bandgap to Eg = 0.91 eV and induced a magnetization of approximately ΔM = 2.0 μ_B_. The induced magnetization can be attributed to the localized orbitals on the water molecules, which corresponded to the gap states shown in Figure 8b. Similar gap states are shown in Figure 8d due to the physisorption of the ozone O_3_ molecule on the same substrate MoS_2_:V_S_. (c) Sample MoSe_2_:V_Mo_: This particular sample was not only ferromagnetic but also further had a half-metallic character, as shown in Figure 8c in the red, shaded curves. The spin-up state had a metallic character, whereas the spin-down state had a semiconducting character with $E_{g}^{down}=0.69\;eV.$ Amazingly, both magnetization and half-metallicity were removed by the molecular adsorption of the oxidizing gases (H_2_O, O_2_, and O_3_). The physisorption processes of H_2_O and O_2_ changed the properties to paramagnetic semiconducting, with $E_{g}^{up}=E_{g}^{down}=0.39\;eV$ and 0.37 eV, respectively. However, the chemisorption of the ozone O3 molecule yielded ferromagnetic metal with a weak magnetization of M = 0.28 μ_B_. Hence, the effects of the atmospheric gases on the electronic and magnetic properties of this sample of MoSe_2_:V_Mo_ were significant. (d) Sample MoSe_2_:V_Se_: Originally, the adsorbent bed was a paramagnetic semiconductor with a bandgap energy of Eg = 1.02 eV. The presence of humidity (H_2_O) affected the bandgap and magnetization. Nonetheless, the physisorption processes of O_2_ and O_3_ had a significant impact on the electronic and magnetic properties. Specifically, these adsorption processes can transform the substrate into a ferromagnetic semiconductor, with magnetization reaching M = 2.0 μ_B_.

## 4. Conclusions

The effects of intrinsic point defects on the electronic structures and magnetic and spintronic properties of TMD MoX_2_ monolayers (X = S and Se) are presented based on both localized and plane-wave base sets using SIESTA and VASP packages, respectively. Among the studied defects, we selected (i) a molybdenum vacancy “V_Mo_”, (ii) a chalcogenide vacancy “V_X_”, and (iii) a di-chalcogenide vacancy “V_X2_”. The vacancies were studied as single-point defects in periodic samples of sizes ranging from 4 × 4 to 8 × 8 PCs. The scaling in the sample size was explored to stimulate the quantum and magnetic effects of the vacancy–vacancy interactions between a real vacancy and its six mirror images. Furthermore, the effects of the adsorption of atmospheric gases (e.g., H_2_O, O_2_, O_3_) on the electronic and magnetic properties were analyzed to access their robustness under ambient conditions. The results can be summarized as follows:(1)In pristine monolayer form, both MoS_2_ and MoSe_2_ were paramagnetic semiconductors with direct bandgaps at the K-point of energies 1.626 eV and 1.460 eV, respectively.(2)The samples of MoS_2_ MLs with any kind of vacancy were found to be paramagnetic.(3)The samples of MoSe_2_ MLs with V_S_ and V_S2_ were also found to be paramagnetic. Only in the case of the “V_Mo_” Mo vacancy could magnetism be introduced into the MoSe_2_ samples to become ferromagnetic. The magnetic moment due to the V_Mo_ was estimated to be 3.99 μ_B_, which remained robust independent of the sample size. However, the samples of sizes 4 × 4 and 5 × 5 PCs were found to exhibit half-metallicity, where the spin-up state changed and became metallic.(4)We investigated the origins of half-metallicity, in particular for the small-sized samples (i.e., 4 × 4 and 5 × 5 PCS) of MoSe_2_ with a single vacancy of V_Mo_. So, we calculated the PDOS, ODOS, and HOMO/LUMO, as well as the eigenstates at the Fermi energy for both the spin-down and spin-up states. Our results showed that half-metallicity can be attributed to the mixing of the $d_{z^{2}}$ and $d_{xy}$ orbitals of the Mo atoms triggered by the FMC interactions.(5)Using a large sample of size 8 × 8 PCs containing MoSe_2_ with four vacancies of V_Mo_, we demonstrated that the ground state was an FMC interaction. The critical distance of the FMC interaction was estimated to be $d_{c}≅16$ Å.(6)The environmental effects due to the interaction of the studied samples with the atmospheric gases (e.g., H_2_O, O_2_, O_3_) were further examined. In MoSe_2_:V_Mo_, the adsorptions of these gases were shown to suppress both the half-metallicity and magnetization. Specifically, the humidity, H_2_O, and O_2_ transformed the substrate into a paramagnetic semiconductor, whereas the ozone O_3_ molecule transformed the substrate into a weak FM metal.

Half-metallicity is a fundamental requirement for the development of spintronic devices. It requires the presence of FMC interactions, which are magnetic-coupling interactions of intermediate strength. However, the effects of the environment (atmospheric gases) must be minimized in order to preserve the half-metallic properties. Understanding the conditions that lead to the appearance/disappearance of half-metallicity is crucial for spintronic device applications.

## Figures and Tables

**Figure 1 nanomaterials-13-01642-f001:**
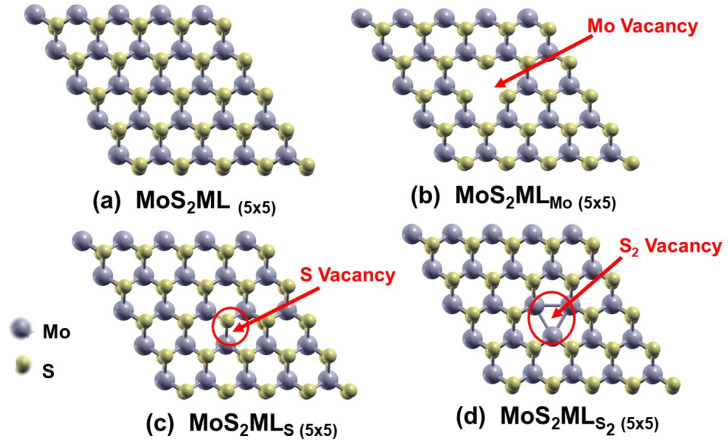
Relaxed geometries of (**a**) pristine MoS_2_ ML (5 × 5), (**b**) MoS_2_ ML (5 × 5) with Mo vacancy, (**c**) MoS_2_ ML (5 × 5) with S vacancy, and (**d**) MoS_2_ ML (5 × 5) with S_2_ vacancy.

**Figure 2 nanomaterials-13-01642-f002:**
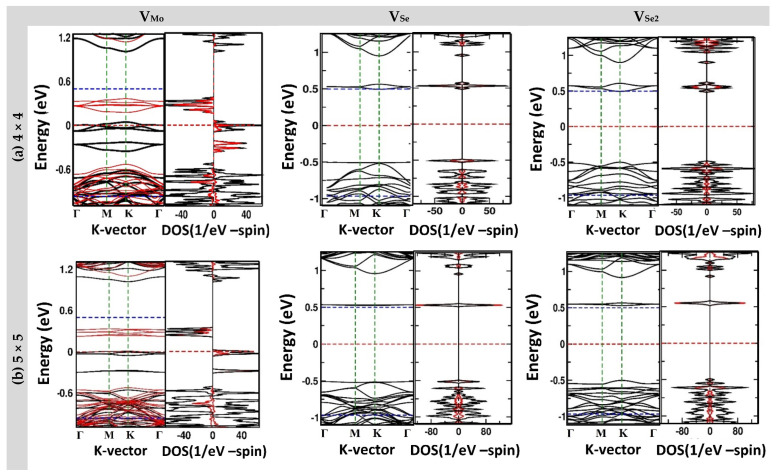
Spin-polarized electronic band structures and density of states (DOS) of vacancy-defected MoSe_2_ ML versus the sample sizes (i.e., 4 × 4, 5 × 5, and 8 × 8 PCs) and type of vacancy: (**a**) V_Mo_; (**b**) V_Se_; and (**c**) V_Se2_. The Fermi energy is chosen as an energy reference (i.e., E_F_ = 0) and is indicated by the red dashed horizontal line. The blue horizontal dashed lines below and above the Fermi level show the valence-band maximum (VBM) and conduction-band minimum (CBM), respectively. For the DOS, red and black represent the contributions of the atoms near the vacancy and total DOS, respectively, whereas for the bands, the red and black curves represent the spin-down and spin-up states, respectively.

**Figure 3 nanomaterials-13-01642-f003:**
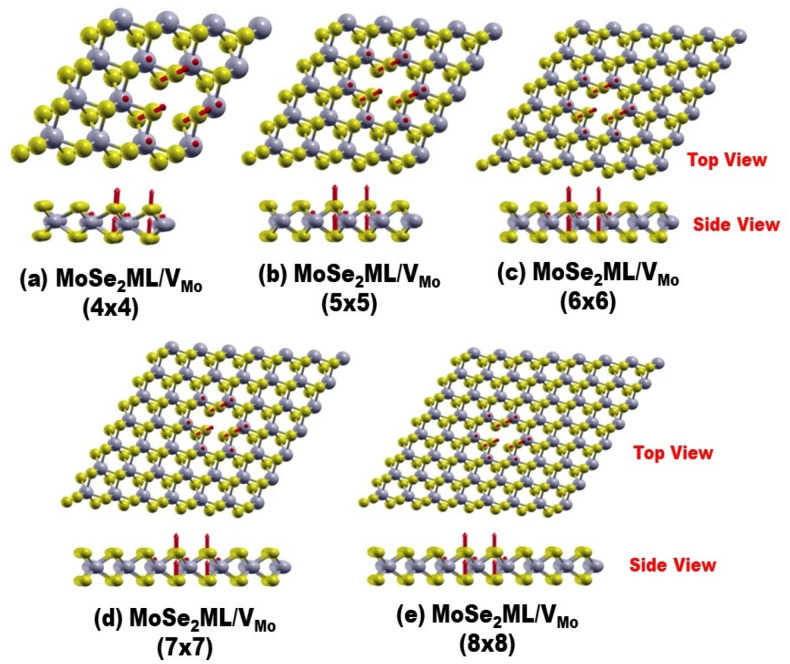
Relaxed samples of various sizes of MoSe_2_ with the Mo vacancy “VMo”. The red arrows indicate the spin vectors, demonstrating the robustness of magnetization irrespective of sample size. Here, 1 defect V_Mo_ in MoSe_2_ ML is shown but on different sample sizes: (**a**) 4 × 4 PCs, (**b**) 5 × 5 PCs, (**c**) 6 × 6 PCs, (**d**) 7 × 7 PCs, and (**e**) 8 × 8 PCs.

**Figure 4 nanomaterials-13-01642-f004:**
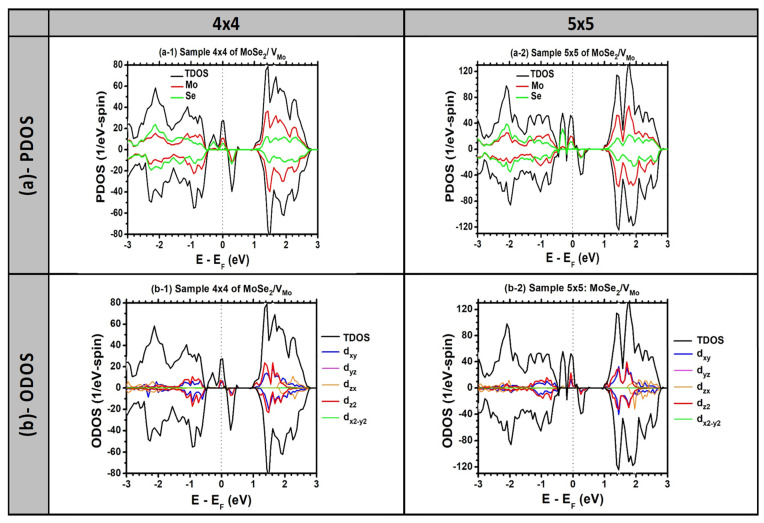
(**a**) Spin-polarized Partial and orbital densities of states “PDOS” and (**b**) Spin-polarized Orbital density of states “ODOS” calculated using the VASP are shown for 2 samples of sizes 4 × 4 and 5 × 5 PCs of MoSe_2_ with Mo vacancy “V_Mo_”. The Fermi level is chosen as an energy reference and is indicated by a vertical dotted line.

**Figure 5 nanomaterials-13-01642-f005:**
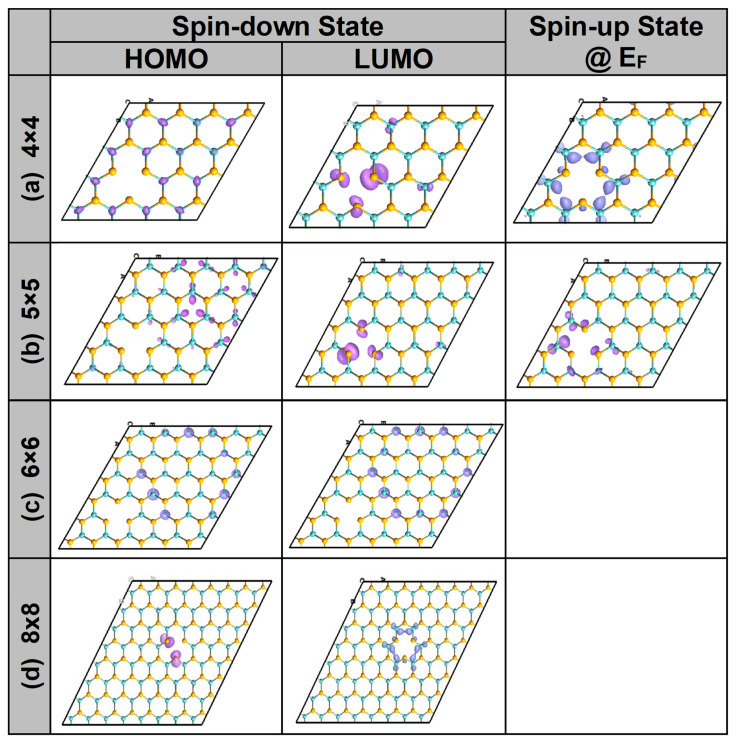
The HOMO and LUMO of the spin-down states and the eigenfunction of the spin-up state at the Fermi level when the latter behaves as a metal are shown for MoSe_2_ with Mo vacancy “V_Mo_” versus 4 sample sizes: (**a**) 4 × 4 PCs, (**b**) 5 × 5 PCs, (**c**) 6 × 6 PCs, and (**d**) 8 × 8 PCs. Mo and Se atoms are denoted by cyan and yellow, respectively. The magnitude of the eigenfunction is represented in decreasing order by colors varying from purple to blue.

**Figure 6 nanomaterials-13-01642-f006:**
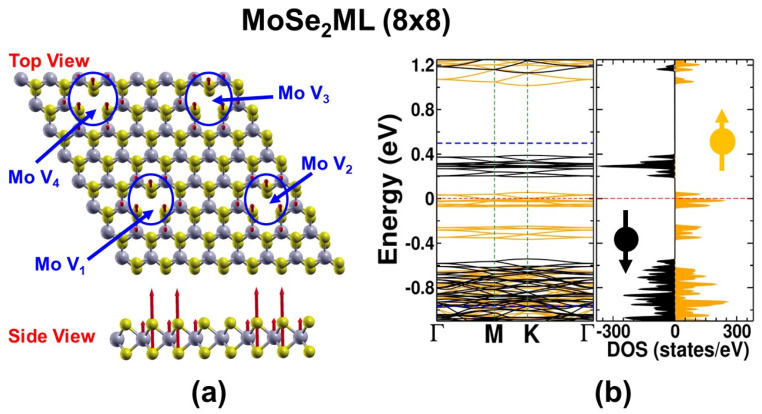
(**a**) Relaxed sample of MoSe_2_ of size 8 × 8 PCs with four Mo vacancies uniformly distributed. This figure shows the ground state to be ferromagnetic. (**b**) The corresponding spin-polarized bands and TDOS show the half-metallic character. Colors: Spin-up states (orange) and spin-down states (black).

**Figure 7 nanomaterials-13-01642-f007:**
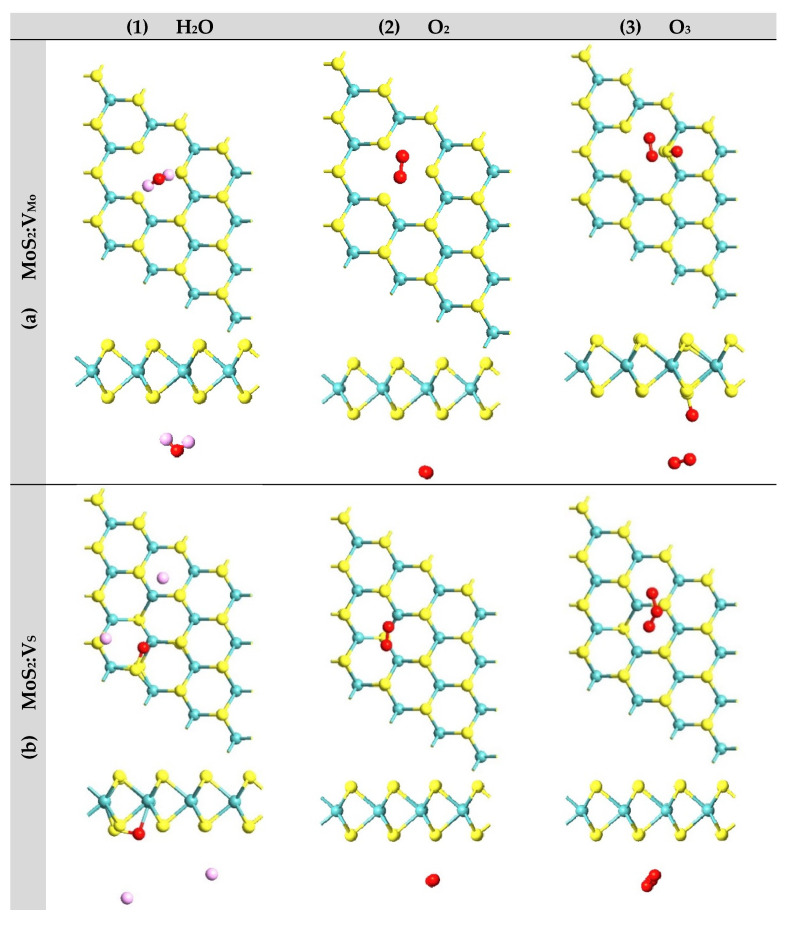
Relaxed atomic structures after the adsorption processes of (H_2_O, O_2_, O_3_) gas molecules on the surfaces of the four defected TMD monolayers: (**a**) MoS_2_:V_Mo_, (**b**) MoS_2_:V_S_, (**c**) MoSe_2_:V_Mo_, and (**d**) MoSe_2_:V_Se_.

**Figure 8 nanomaterials-13-01642-f008:**
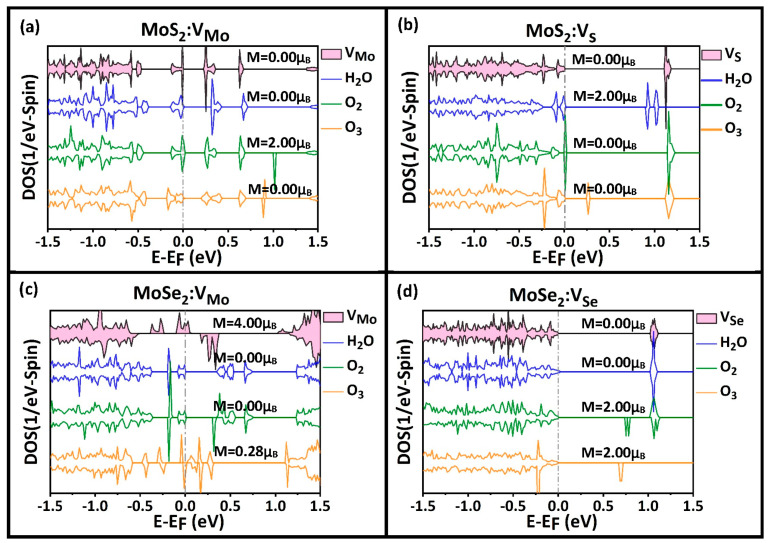
Spin˗polarized TDOS before and after the adsorption of 3 gases (H_2_O, O_2_, O_3_) on the surfaces of four defected TMD monolayers: (**a**) MoS_2_:V_Mo_, (**b**) MoS_2_:V_S_, (**c**) MoSe_2_:V_Mo_, and (**d**) MoSe_2_:V_Se_. The Fermi level is chosen as an energy reference (E_F_ = 0) and the values of magnetization are shown.

**Table 1 nanomaterials-13-01642-t001:** Spin-polarized bandgap energy and magnetization before and after gas adsorption (H_2_O, O_2_, O_3_) on defected TMD substrates. Spin-up and spin-down are indicated by an arrow up (↑) and an arrow down (↓), respectively.

	Before Gas Adsorption		After Gas Adsorption					
Substrate	E_g_ (eV)	M (μ_B_)	H_2_O		O_2_		O_3_	
			E_g_ (eV)	M (μ_B_)	E_g_ (eV)	M (μ_B_)	E_g_ (eV)	M (μ_B_)
MoS_2_:V_Mo_	0.234(↑↓)	0.00	0.31(↑) 0.31(↓)	0.00	0.25(↑) 0.25(↓)	2.00	0.23(↑) 0.23(↓)	0.00
MoS_2_:V_S_	1.10(↑↓)	0.00	0.91(↑) 0.91(↓)	2.00	1.16(↑) 1.16(↓)	0.00	0.25(↑) 0.25(↓)	0.00
MoSe_2_:V_Mo_	0.00(↑) 0.69(↓)	4.00	0.39(↑) 0.39(↓)	0.00	0.37(↑) 0.31(↓)	0.00	0.07(↑) 0.00(↓)	0.28
MoSe_2_:V_Se_	1.02(↑↓)	0.00	1.00(↑) 1.00(↓)	0.00	1.02(↑) 0.74(↓)	2.00	1.51(↑) 0.67(↓)	2.00

**Table 2 nanomaterials-13-01642-t002:** Adsorption energy, charge transfer, and molecule–substrate distance of adsorption of H_2_O, O_2_, and O_3_ molecules on defected TMD substrates.

	H_2_O			O_2_			O_3_		
	E_ads_ (eV)	∆Q (e)	Distance (Å)	E_ads_ (eV)	∆Q (e)	Distance (Å)	E_ads_ (eV)	∆Q (e)	Distance (Å)
MoS_2_:V_Mo_	−0.12	−0.15	d *(H-Se) = 2.50	0.01	−0.03	d(O-S) = 3.61	−2.56	−0.69	b(O-Se) = 1.50
MoS_2_:V_S_	6.88 (H2O) −6.786 (O)	−0.98	b *(O-Mo) = 2.13 b(O-S) = 1.60	−0.01	−0.09	d(O-S) = 2.92	−0.09	−0.19	d(O-S) = 2.67
MoSe_2_:V_Mo_	−0.19	−0.20	d(H-Se) = 2.57	−0.10	−0.13	d(O-Se) = 3.1	−1.57	−1.29	b(O-Se) = 1.69
MoSe_2_:V_Se_	−0.29	−0.37	d(H-Se) = 0.98	−0.18	−0.26	d(O-Se) = 1.89	−5.53	−0.98	b(O-Mo) = 2.07

* d = distance, b = bond length.

## Data Availability

The datasets used and/or analyzed during the current study are available from the corresponding author on reasonable request.

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
