# Peer review of "Ferromagnetism in Defected TMD (MoX2, X = S, Se) Monolayer and Its Sustainability under O2, O3, and H2O Gas Exposure: DFT Study"

_nanomaterials, 2023, doi:10.3390/nano13101642_

Round 1

Reviewer 1 Report

The manuscript by Devi et al is devoted to first principles study of MoX2 compounds (x = S, Se) and their properties with respect to different defects.

First of all I would say that manuscript is too long, it is very difficult to read it as Introduction section more than 4 pages. There are lots of information that is not really needed here. 

Authors should concentrate on the main achievement related to the topic of the urrent study, otherwise it is not understandable what really authors want to say by this long introduction. 

Why do we need to read about you studied system in the introduction and the plan of your study. There are no need to write this information in the introduction. Please read other papers that have already published in this journal to understand the required style of writing.

The same story is for computational details section. A lot of information but almost no details. 

The most important question related to Hubbard U value. Authors did not provide any justifications why they choose 5 eV for all the structures? Did they provide any tests of what U-value is more appropriate? This should be clarified.

Figure 1 is useless, there is a well-known information about electronic structure of MoS2 and MoSe2 which is not necessary to show in the results section as separate figure and separate paragraph.

Did the authors compute formation energy of the defects? Did they check the stability of studied defects? Maybe they are not stable at all and cannot be formed in the monolayers?

Authors compared the binding energies with references made by the authors. It would be better to compare your results with results of other authors on this topic.

What about the surface energy? Did the authors calculate surface energies of various structures they considered?

The quality of figure 2 is extremely poor. There is no possibility to understand what is shown there. Should be redrawn.

Another question is related  to Figure 2(c). Why do you have such changes on the behavior of conduction band in comparison with 4x4, 5x5, 8x8? Why band structure of 6x6 significantly changes from others? This should be mistake.

There are many other things that should be addressed, but the paper is too long for careful reading. 

So, I recommend to significantly change the manuscript, make it more bulky, more solid, to show the most relevant and important information describing your results rather than such long story...

Author Response

  • The manuscript by Devi et al is devoted to the first principles study of MoX2 compounds (x = S, Se) and their properties with respect to different defects.

First of all, I would say that manuscript is too long, it is very difficult to read it as the Introduction section is more than 4 pages. There are lots of information that is not really needed here. 

Authors should concentrate on the main achievement related to the topic of the current study, otherwise, it is not understandable what really authors want to say by this long introduction.

Why do we need to read about your studied system in the introduction and the plan of your study? There is no need to write this information in the introduction. Please read other papers that have already been published in this journal to understand the required style of writing.

Response: We thank the Referee for guidance and positive comments. Indeed, we have put great effort to shorten the Introduction section and making it concise and precise. We only elaborated on the points of relevance to the current research article.

  • The same story is for the computational details section. A lot of information but almost no details. 

Response: Thanks to the Referee again. We have shortened section 2 as well. Nevertheless, we have added formulas relevant to our calculations to comply with the recommendations of both Referees.

  • The most important question related to Hubbard U value. The authors did not provide any justifications for why they choose 5 eV for all the structures. Did they provide any tests of what U-value is more appropriate? This should be clarified.

Response: We agree with the Referee about the importance of the on-site U Hubbard parameter in case of inclusion of transition metal elements in the calculations, such as Molybdenum (Mo) in our TMDs MoX2 (with X = S, Se). Many authors in the literature did the optimization of parameter U and reported that higher values are usually needed to achieve the ground-state atomic structures, but it should not also be too large in order to remain in the zone of correctness vis-à-vis the electronic structures (see Ref.1 below). In our presented work, the optimization of U Hubbard was not in the scope of our investigation. We have used the default values existing in both VASP and ATK packages. That value was about U = 4.5 eV and was also successfully used in our previous work (Ref.1-2 below + Ref.# 26 in the revised MS). A brief justification is added in the text of MS in section 2.

Ref.1: Mann, G.W.; Lee, K.; Cococcioni, M.; Smit, B.; Neaton, J.B. First-principles Hubbard U approach for small molecule binding in metal-organic frameworks. J. Chem. Phys. 2016, 144, 174104.

Ref.2: Tolba, S.A.; Gameel, K.M.; Ali, B.A.; Amossalami, H.A.; Allam, N.K. The DFT+U: Approaches, Accuracy, and Applications. “Density Functional Calculations: Recent Progresses of Theory and Applications”, Ed. Gang Yang (Intech Open, 2018, ISBN: 971-1-83881-327-7).

  • Figure 1 is useless, there is well-known information about the electronic structure of MoS2 and MoSe2 which is not necessary to show in the results section as a separate figure and separate paragraph.

Response: We agree with the Referee. This Figure and its corresponding text were omitted completely in the revised version of the manuscript.

  • Did the authors compute the formation energy of the defects? Did they check the stability of the studied defects? Maybe they are not stable at all and cannot be formed in the monolayers.

Response: We thank the Referee for this excellent remark. Yes, indeed, we have elaborated in section 2 on how to calculate the formation energy of a defect, such as a vacancy. We have shown our results in cases of Mo-vacancy in both MoS2 and MoSe2 MLs to be = -1.341 eV and -1.095 eV, to be consistent with results of Ding and coworkers (Ref. 27 in new MS), of values  = -1.420 eV and -1.210 eV, respectively. Based on the experimental reports, our theoretical results of average binding energies and formation energies of defects are all in favor of the thermodynamic stabilities of our defects. We have added discussions on this issue in sub-section 3.1.

  • The authors compared the binding energies with references made by the authors. It would be better to compare your results with the results of other authors on this topic.

Response: We thank the Referee for this critical comment. Yes, indeed, we have added two new references (i.e., Refs # 27 and 28 in our revised MS) to compare our results of binding energies in cases of pristine MoS2 and MoSe2 MLs. Tables S1 and S2 as well as the text in MS is updated accordingly.

  • What about surface energy? Did the authors calculate the surface energies of various structures they considered?

Response: We apologize that we kept this issue out of the scope of the present work.

  • The quality of Figure 2 is extremely poor. There is no possibility to understand what is shown there. Should be redrawn.

Another question is related to Figure 2(c). Why do you have such changes in the behavior of the conduction band in comparison with 4x4, 5x5, and 8x8? Why band structure of 6x6 significantly changes from others? This should be a mistake.

Response: Actually, we recall that this Figure was number 3 in the old MS and is now number 2 in the revised version of MS. As a matter of fact, we did some effort to enhance this Figure. We eliminated completely the panel containing the results of sample 6x6 PCs to get rid of confusion and enhance the clarity of the Figure. We hope that now, this Figure looks better and complies with the Referee’s expectations.

  • There are many other things that should be addressed, but the paper is too long for careful reading. 

So, I recommend significantly changing the manuscript, making it bulkier, more solid, to show the most relevant and important information describing your results rather than such a long story...

Response: We thank the Referee for the guidance and relevant comments and recommendations. We did benefit a great deal from those comments in improving and enhancing the presentation of our results in this paper. We truncated many parts in all sections, including sections 1, 2, and 3. We hope that the current revised manuscript would please the Referees. 

Reviewer 2 Report

Authors of this work have tried to examine the adsorption of atmospheric gases (H2O, O2, O3) on MoX2 systems and the way these foreign species impact the electronic and magnetic properties. Although the motive is not discouraging, this ms needs serious work.

1.      This article needs significant write up. The introduction section should not be four pages long. It should be reduced to a maximum of two or three paragraphs. Otherwise, it will make the reader tiring.

2.      It is not clear why did the authors use two software (VASP and SIESTA) for their calculations?

3.      Why was atomic relaxation performed using SIESTA? Why cannot it be done with VASP?

4.      Equations should be given that can describe how vacancy formation energy was calculated?

5.      Fig. 3 is blurry, and I cannot read it.

6.      The paper is too long, and the science discussed is not compact and solid.

Clearly, I suggest the authors to prepare the MS based on my comments 1-6, and resubmit.

Author Response

Authors of this work have tried to examine the adsorption of atmospheric gases (H2O, O2, O3) on MoX2 systems and the way these foreign species impact the electronic and magnetic properties. Although the motive is not discouraging, this MS needs serious work.

  1. This article needs a significant write-up. The introduction section should not be four pages long. It should be reduced to a maximum of two or three paragraphs. Otherwise, it will make the reader tired.

Response: Thanks to the Referee for the positive comments and guidance. Indeed, we truncated the Introduction section and made it concise and precise. Now, it is 1 and a half pages and mainly comprises about 4 paragraphs. We hope that this new length of introduction should be okay with the Referee.

  1. It is not clear why did the authors use two software (VASP and SIESTA) for their calculations?

Response: We thank the Referee for the critical observation. In section 2, we have elaborated on the reason why both VASP and SIESTA were used in our investigation. The subject of our study is TMD MLs, which are composed of 1/3 of a total number of atoms to be transition metal (Mo) atoms. Meanwhile, an investigation of large systems comprising about 200 atoms is needed and for that reason, localized-basis set methods, such as “SIESTA” would be suitable to handle large systems. On the other hand, we used VASP to benchmark the results, especially in small-sized samples (4x4 and 5x5 PCs). Meanwhile, half-metallicity takes place in small samples, where accuracy is needed and VASP is popular for its reliability to get trustworthy results. So, both methods are complementary to each other in our present investigation.

  1. Why was atomic relaxation performed using SIESTA? Why cannot it be done with VASP?

Response: In both methods, the atomic relaxations must be done first before any other calculation. As we mentioned above, the calculations of most of the big samples are carried out using SIESTA. Benchmarking is done always with VASP, especially for small-sized samples, and the properties of half-metallicity there.

  1. Equations should be given that can describe how vacancy formation energy was calculated.

Response: Yes, we agree with the Referee. In the revised manuscript, we have explicitly given the formulas needed in the calculations of binding, formation, and adsorption energies in section 2.

  1. 3 is blurry, and I cannot read it.

Response: Thanks to the Referee for this comment. Yes, we have enhanced the old Figure 3 (Now, it is numbered Figure 2 in the revised version of MS).

  1. The paper is too long, and the science discussed is not compact and solid.

Response: Thanks to the Referee for the positive comment. Indeed, we did a lot of effort to truncate many parts from sections 1, 2, and 3. We hope that the new form of the revised manuscript would please the Referee.

Clearly, I suggest the authors prepare the MS based on my comments 1-6 and resubmit.

Response: Thanks to the Referee for the positive comments and guidance to us to improve and enhance the presentation of our results. We hope that the newly revised manuscript would please the Referees.

Round 2

Reviewer 1 Report

Authors addressed all my issues from revision. I think that manuscript now is more or less suitable for publication while careful proofreading is still required as there are many incoherent sentences etc.

Author Response

Response: Thanks to the Referee for the good impression. As a matter of fact, we have done extensive English revision on the whole manuscript and we incorporated many corrections. Our efforts targeted the reduction of incoherent sentences. I hope that the new version of the manuscript is much improved and would please the Referee.

Reviewer 2 Report

Authors of this work have revised their ms based on my suggestions. It may now be acceptable for possible publication.

Author Response

We thank the Referee for the nice impression.